# What the Patient Thinks and What the Patient Does: Placebo, Nocebo, and Therapy Adherence in Ulcerative Colitis

**DOI:** 10.3390/jcm14124351

**Published:** 2025-06-18

**Authors:** Emanuela Ribichini, Giorgia Burrelli Scotti, Simone Di Cola, Giulia Scalese, Carola Severi, Filippo Vernia

**Affiliations:** 1Department of Translational and Precision Medicine, Sapienza University, 00185 Rome, Italy; giorgia.burrelliscotti@uniroma1.it (G.B.S.); simone.dicola@uniroma1.it (S.D.C.); giulia.scalese@uniroma1.it (G.S.); carola.severi@uniroma1.it (C.S.); 2Department of Life, Health, and Environmental Sciences, Division of Gastroenterology, Hepatology, and Nutrition, University of L’Aquila, Piazza S. Tommasi, 1- Coppito, 67100 L’Aquila, Italy; filippo.vernia1@gmail.com

**Keywords:** adherence, placebo, nocebo, inflammatory bowel disease, IBD, ulcerative colitis, UC

## Abstract

Patients’ attitude toward therapy and adherence to treatment are central in determining the long-term outcomes of medical treatment in ulcerative colitis. A complex interplay of differing factors modulates the likelihood of persisting in or discontinuing treatment, including patients’ beliefs and concerns about adverse effects of drugs, as well as the interactions with medical staff. Emotional attitude and expectancies are reflected in the so-called placebo and nocebo effects which influence patients’ choices to adhere to or discontinue treatment. They represent important confounding factors in clinical trials and are amplified when the evaluation relies on patient-reported outcomes more than on objective measurements. The therapeutic gain related to placebo effects is likely also relevant in day-to-day practice, but few data are available. The aim of the present narrative review is to provide critical insight into the adherence to therapy in ulcerative colitis and its interaction with the emotional component of the effects of therapy, resulting in the placebo/nocebo effects. Understanding the mechanisms underlying patient behavior may help identify the most appropriate therapeutic approach and treatment schedule to optimize adherence and outcomes in individual patients with UC.

## 1. Introduction

Ulcerative colitis (UC) is a chronic inflammatory bowel disease (IBD) characterized by the continuous mucosal inflammation of the colon, typically starting in the rectum and extending proximally. Common symptoms include bloody diarrhea, abdominal pain, urgency, and tenesmus, with varying degrees of severity and clinical course.

Current treatment strategies for UC are based on disease extent and severity, ranging from topical or oral 5-aminosalicylates (5-ASA) in mild cases, to corticosteroids, immunosuppressants (e.g., thiopurines), and advanced therapies such as biologics and small molecules in moderate-to-severe disease [1].

The primary therapeutic goal is to achieve and maintain clinical and endoscopic remission, thereby improving quality of life and preventing complications [2]. The effectiveness of these therapies, however, can be significantly influenced by factors such as patient adherence, placebo and nocebo effects, and individual expectations—elements increasingly recognized as pivotal in both clinical trials and real-world practice. Consequently, the outcome of medical treatment results from the interplay of multiple factors. While some are inherent to the disease itself or the pharmacological properties of the drug—and thus not easily modifiable—others can be positively influenced. Notably, interventions aimed at improving patient attitudes toward therapy and optimizing therapeutic strategies may yield substantial benefits. This is especially true for chronic, relapsing–remitting conditions such as UC, which often require long-term or lifelong management.

Therapeutic adherence is a keystone in the management of UC, especially in remission, as maintenance therapy with 5-ASA prevents relapses, favors steroid-free remission, and may reduce the long-term risk of colorectal cancer [3,4]. Nonetheless, widely differing figures for adherence/non-adherence have been reported in different studies due to differences in definition, the tools used for quantifying its prevalence, the composition of patient population, and the geographical area in which the study has been carried out. The scenario partially changed in the biologic era as the disadvantage of inadequate compliance to prescribed drug therapy in non-active UC is likely less marked than in the past [5].

Less attention has been given to the emotional attitude and expectancies in UC patients, resulting in the favorable or unfavorable effects of drugs, although their active ingredients lack this potential. Indeed, the so-called placebo (PE) and nocebo effect (NE) [6] are factors markedly influencing patients’ choices to adhere to or discontinue treatment. Their identification is central for deciding which therapeutic strategy each patient is more likely to accept and optimize outcomes.

The aim of the present narrative review is to provide critical insights into the adherence to therapy in UC and its interaction with the emotional component of the effects of therapy, resulting in the PE/NE.

## 2. Adherence to Treatment

The term “adherence” refers to the extent to which a person’s behavior—taking medication, following a diet, and/or making lifestyle changes—corresponds to agreed recommendations from a health care provider. It is primarily based on the cooperation between physicians and patients. The term is generally preferred to “compliance”, i.e., the patient’s passive act of following instructions, consisting of simple obedience to prescriptions [7,8].

Conversely, non-adherence (NA) is a primary cause of ineffective treatment. It worsens the course of the disease and increases treatment costs (drug implementation, increased morbidity, and mortality) [9]. NA has been reported in 30–60% of patients affected by inflammatory bowel disease (IBD) and is similar in UC and Crohn’s disease (30–45%) [10,11,12]. However, the recorded prevalence of NA depends on the method used for the assessment, i.e., patient-reported outcomes, pharmacy refill records, or drug metabolite levels [13]. For example, Lachaine et al. [14] used refill data from the Régie de l’assurance maladie du Québec (RAMQ) database in Canada to objectively assess adherence, while D’Incà et al. [12] in Italy relied on patient interviews (self-report). In Spain, Ballester et al. [5] utilized electronic management systems, offering more automated and real-time adherence tracking. The issue is complex and involves many factors, including disease extent and duration, the cost of medications, the fear of adverse effects, and the patient–physician relationship (trust in the attending physician, adequacy of information). When focusing attention on NA in the clinical setting, asking the patient “how often you skip doses” instead of “do you regularly assume the prescribed dose” significantly increases the identification of non-compliant patients [15]. Irrespective of the method used for assessing NA, individual psychosocial variables such as single status, male gender, full-time employment, high depression scores, and low quality of life are all negative predictors. The above-mentioned factors vary in different countries in relation to healthcare systems and economic and cultural background [16]. For instance, Henriques et al. [17] documented cultural and structural influences on complementary and alternative medicine (CAM) use and adherence behaviors in Brazil, while Almakadma et al. [18] studied these relationships in the Saudi Arabian population, highlighting differences related to trust in conventional medicine and access to healthcare. Vavricka et al.’s UCandME European survey also emphasized regional variability in adherence linked to health literacy and system support [11].

### 2.1. Adherence to Treatment in UC

**Oral aminosalycilates:** Adherence to aminosalycilates in community-based studies is 28–40%. These figures are lower than those reported in clinical trials at around 80% [12,14,19,20]. Patient selection and close monitoring are likely responsible for the difference. Lachaine et al. reported higher adherence to multi-matrix (MMX)–mesalamine treatment than to any other type of oral mesalamine (40.9%, *p* < 0.001) due to once-daily administration [14]. Subsequently, several studies in active UC showed that other mesalamine formulations may also be administered once daily without affecting the remission rates and are associated with higher satisfaction rates compared to divided doses [20].

**Topical aminosalycilates:** Topical therapy with foams, suppositories, or enemas, being less easily accepted due to embarrassment and discomfort, is more likely associated with NA than oral therapy (68% vs. 40%, *p* = 0.001) [12]. Using a medication possession ratio criterion, Boyle et al. reported 71% NA in patients prescribed with rectal mesalamine [21].

**Immunosuppressants:** The adherence rate to thiopurines in UC patients ranges from 63 to 78% and it is higher than that to oral mesalamine [22,23,24]. The rate of NA to immunosuppressants is not modified by the concomitant use of biological agents [24]. Disease awareness and the chronic activity of the disease are likely responsible for these favorable findings [22].

**Biologics:** The overall reported risk of inadequate adherence to biologics involves 19.5% of UC patients and is lower than the risk of inadequate adherence to oral 5-aminosalicylates (odds ratio (OR) 4.1) and thiopurines (OR 1.7). The proportion of inadequate adherence is significantly higher for subcutaneous biologics than intravenous ones (28.5% vs. 7.7%; OR 4.8) [23]. The difference results from the medical setting, which is either hospital-based or home-based and self-administered. Longer disease duration and more regular contact with gastroenterologists are protective factors against NA. This may be clinically relevant as low adherence is associated with an increased risk of hospitalization [25].

Optimal adherence to Tofacitinib, an oral small molecule inhibiting Janus kinase (JAK), has been reported. Therapy covered 95.7% of days in UC patients. This was likely due to the oral route of administration and the high expectancy in patients who had previously failed treatment with one or more biologic agents [26].

**Oral steroids:** Data on adherence to steroids is limited and partially conflicting. Cervený et al. [27] using a 30-item interview found an NA rate of 40% in UC patients on systemic steroids. Results were strongly influenced by the occurrence of adverse effects, including morphological changes, neuropsychological disorders, and weight gain [27]. Instead, Severs et al. reported a 9.8% rate, using the visual analogue scale > 80 [28]. A 25% NA rate was reported when budesonide–MMX treatment was prescribed at the dose of one pill once a day. Quick improvement in patients with mild disease [29] and the low bioavailability of the steroid, resulting in minimal systemic side effects, account for the result.

### 2.2. Adherence in Special Groups of UC Patients

**Pregnancy:** Watanabe et al. [30] investigated the incidence of NA in 68 pregnant UC women using a patient self-reporting system. Adherence to biologics and steroids exceeded 80% during pregnancy. Adherence to mesalamine and thiopurines drops in the first trimester due to the fear of harmful effects on the fetus, despite being good prior to pregnancy. This behavior is often underestimated by physicians. Interestingly, adherence improves in the third trimester, thanks to effective counselling provided by physicians during the follow-up visits after conception. As NA represents an independent risk factor for relapse [odds ratio (OR) 7.7, *p* = 0.038], unfavorable pregnancy outcome may result (OR 8.4, *p* = 0.023) [30].

Only 44% of IBD women breastfed their infant due to physician recommendation, fear of medication interactions, and personal choice. Most lactating women stop therapy, increasing the risk of disease relapse [31].

**Pediatric population:** Factors influencing adherence in pediatric chronic disease are more complex than in the adult population. Both patients and parents must be considered, as parents usually provide medication for their children. Considering IBD-specific medications, 48% of children and 38% of parents reported strict adherence to all prescribed medications, with high parent–child concordance [32]. Risk factors associated with worse adherence include family dysfunction, poor stress-coping strategies, and overall worse behavioral/emotional functioning. Medication adherence is even more complicated in adolescence as patients in this age group are willing to take decisions autonomously [32], irrespective of the advice of physicians and parents. A retrospective study in UK carried out in 607 UC adolescents and young adults (aged 10–24 years) reported that oral 5-aminosalicylate maintenance treatment was discontinued within 1 month from prescription in one-quarter of patients and was discontinued in two-thirds of them within the first year. Adherence in this retrospective study was lower in young adults than in adolescents (69% in those aged 18–24 years versus 80% in those aged 10–14 years). The declining adherence to mesalamine over time was highly predictive of the need for treatment escalation [33]. Conversely, early corticosteroid use due to acute flares lowers the risk of NA [34].

### 2.3. Adherence to Other Clinical Recommendations

**Surveillance colonoscopy:** Friedman et al. documented that the adherence rate to colorectal cancer (CRC) screening is low [35]. Only 25.5% of IBD patients (half of them affected by UC) underwent surveillance colonoscopies every 3 years. Self-reported adherence was consistently higher than chart-documented adherence, as patients often lose track of time and delay the subsequent procedure longer than anticipated (unintentional NA). Additional significant factors favoring NA included logistics, health perception, procedure-related stress, and problems in one’s working and personal life that caused procedural problems. The most frequently reported primary reason was bowel preparation. The detection of dysplasia was unrelated to low adherence, but the absolute number of patients in the series was too small to provide reliable conclusions [36]. The recent Sars-CoV2 pandemic also represented an additional cause of NA and involved drug therapy as well as elective endoscopic procedures [36].

**Vaccines:** Adherence to vaccination is less than optimal in patients with UC, more so considering the increased risk of infections resulting from chronic disease and the use of immunosuppressants [37]. This mainly results from the fear of side effects, poor adherence to medical treatment, and single status. Conversely, treatment with specific IBD drugs, such as as immunosuppressants and/or biological agents, favored adherence [38].

Because of disease severity, information campaigns and mediatic pressure SARS-Cov-2 vaccination rates represented a positive exception. However, the number of subjects undergoing the third, fourth, or subsequent doses progressively declined [39,40]. Thus, the suboptimal IBD vaccination rates against other infectious diseases before the COVID-19 pandemic likely resulted from physicians’ low attention to the issue and inadequate patient awareness. Indeed, several meta-analyses confirm that proactive physician intervention increases the vaccination rates in IBD [41].

**Diet:** Some epidemiological data link diet and disease occurrence but current evidence does not support the role of food-based therapies in inducing or maintaining clinical remission in UC [42]. Dietary intervention restricting FODMAPs is associated with symptom relief. The adequate intake of calories, vitamins, and micronutrients is of prime importance, especially considering that up to 77% of patients restrict food groups, including vegetables, fruit, and milk derivatives. The opinion that lactose-containing food and fiber worsens diarrhea prompts the restriction of dairy products and green vegetables, resulting in inadequate calcium, vitamin D, and K intake [43]. The role of negative expectations and the NE will be examined in detail in the following section.

All things considered, following nutritional counselling and education, UC patients willingly adhere to Mediterranean diet [44], which is considered a useful addition to front-line pharmacological treatment.

**Self-medication:** About 15% of IBD patients use corticosteroids without medical prescription following diagnosis, seeking the quick relief of symptoms without consulting the attending physician [45]. Self-medication is significantly more common in patients treated in general gastroenterology units than in those treated in dedicated IBD centers. NA and self-medication are more frequent in patients not attending regular follow-up visits or presenting with more frequent disease flares [46]. The prevalence of self-medication with analgesics is high (49.8%) in UC patients [47], and paracetamol (45.2%) and metamizole (21.2%) are the most frequently used drugs. Less than 5% patients use other nonsteroidal anti-inflammatory drugs or opioids. The need for quick symptom relief is the most frequently reported reason for self-medication, and these drugs had often been previously prescribed by the attending physician.

Half of IBD patients use some form of medication or practice not considered as conventional medicine, defined as the use of CAM to control symptoms and manage disease. This behavior is influenced by cultural context and healthcare access. For instance, Henriques et al. [17] found high CAM use among IBD patients in Brazil, associated with regional beliefs and limited trust in conventional care. Similarly, in Saudi Arabia, Almakadma et al. [18] reported CAM usage as a compensatory strategy where health literacy or adherence was suboptimal. CAM includes herbal therapies, dietary supplements, probiotics, and other physical or spiritual practices affecting the mind and body. Patients with low adherence to conventional therapies are more likely to be users of complementary and alternative medication [17,18].

### 2.4. Impact of NA and Strategies to Improve Adherence

A proportion of patients with UC experience periods of unemployment or inability to work, and the subsequent loss of earnings, at some point during their working life [48]. Disability is strongly related to non-adherence to medication [49]. NA increases the risk of symptomatic relapses and healthcare costs and adversely affects the quality of life. Patients with quiescent UC who fail to adhere to their prescribed 5-aminosalicylate regimen have a 61% chance of relapse compared to an 11% chance in adherent patients (*p* = 0.001) [19] and disease activity is significantly associated with depression [50].

Relapses are associated with a two- to three-fold cost increase in the UK for those who do not require hospitalization and a 20-fold increase for hospitalized cases compared to quiescent cases [51]. This underlines the relevance of adherence, both for patient and health-care systems, and of strategies aimed at improving adherence. The systematic review of Gohil compared four intervention categories: educational (pharmacist counseling, use of leaflets, or pill cards), multicomponent (behavior management, guided problem solving, self-management training, and disease education), behavioral (reminder text messages), and cognitive behavioral (problem solving skills training). Multicomponent interventions proved the most effective in IBD patients, but all intervention strategies led to favorable results [52]. Reviewing the patient’s disease and therapeutic history, and identifying which treatments were—or were not—effective in the past provide predictive information on medication NA behavior. Overall, a tailored approach providing individualized intervention, including dose simplification, pillboxes, visual medication reminders, and mobile phone alerts, improves adherence (44%) [53]. In clinical settings where time-consuming approaches to identifying NA are not always feasible, simply asking “how often you do not take all prescribed pills?” instead of “do you regularly take all prescribed pills” will identify two-thirds of NA patients requiring implement active intervention [15].

Despite the decreasing incidence of the colorectal neoplasia, patients with UC are still at an increased risk of developing colorectal cancer (CRC) compared to the general population [54]. The protective role of 5-ASA for CRC is well documented, and the risk of developing CRC is reduced in patients who regularly take aminosalicylates compared those who do not (adjusted OR 0.6) [4,55]. This further highlights the relevance of strategies focused on optimizing adherence to therapy in UC patients, even in the biologic era.

## 3. Placebo Effect and Placebo Response

The placebo effect (PE) is defined as any improvement in symptoms or signs following a physically inert intervention and is clinically relevant, both within clinical trials and in ordinary clinical settings. The term, documented since the 18th century, originates from the Latin placebo, literally translated to “I shall please”. It reflects the idea that the patient will find the treatment agreeable or beneficial and is considered one of the oldest and most effective therapeutic treatments known to humanity. Closely related, the placebo response (PR) is the overall clinical improvement seen in patients receiving placebo. It includes the psychological and neurobiological effects of PE, as well as non-specific factors like natural symptom fluctuation, regression to the mean, and improved care. While PE refers to the underlying mechanism, PR reflects the measurable outcome, which is especially relevant in clinical trials when assessing treatment efficacy.

Blinded or double-blind studies showed that patients treated with inactive treatment (placebo) sometimes exhibit better therapeutic outcomes compared to those who receive active medications [56]. Minimizing the placebo rates in randomized controlled trials (RCTs) is crucial to detect treatment differences between interventions. High placebo rates have been observed in clinical trials in UC, using both traditional drugs (steroids, 5-ASA and immunomodulators) and biologics [57]. The therapeutic landscape significantly expanded following the introduction of new biologics and small molecules that target at least five distinct mechanisms [58]. This improved the possibility of tailoring treatments to the needs of individual patients but rendered the choice more complex. The identification and control of factors influencing placebo rates is crucial for designing effective RCTs aimed at detecting true differences between treatment and control groups, or between different biologics.

Placebo in RCTs: Participants in RCTs are randomly assigned to receive either active treatment or dummies. This minimizes bias and ensures that the observed effects are due to the active therapy itself rather than other factors. However patient expectations, inclusion criteria, patient–provider interaction, baseline disease activity, and psychosocial factors, influence the occurrence of PE. Evidence from multiple therapeutic areas suggests that general trial design features attenuate or amplify PR and remission rate (Table 1) [59,60,61,62,63].

This may help to distinguish the true efficacy of the intervention from the PR.

Several factors influence the PR and remission rates in UC [59]. Studies that required a minimum rectal bleeding score for eligibility showed higher PR rates compared to those that did not. Conversely, trials enrolling patients with more severe endoscopic disease at entry had lower PR and remission rates, underscoring the critical importance of the objective evaluation of disease activity before qualifying patients for enrolment. Similarly, PR was higher in patients with normal CRP, whose symptoms at enrolment were functional rather than related to active UC [62]. This suggests that disease activity is not the only determinant of PE and that the tool used for its assessment (i.e., the ulcerative colitis disease activity index (UCDAI) [64] and the Mayo Clinic Score [65] are of prime importance.

A Cochrane Systematic Review evaluated the PR and remission rates in 61 UC trials (58 induction and 12 maintenance phases), including over 6500 patients randomized to placebo [61]. Placebo-controlled RCTs of adult patients with UC, treated with corticosteroids, aminosalicylates (5-ASA), immunosuppressives, or biologics, were included, but those using antibiotics, probiotics, or complimentary therapies were not. To ensure consistency in measuring disease severity, the Cochrane meta-analysis only included those studies using UCDAI or Mayo Score for patient enrolment and outcome assessment. Similar evidence emerged from papers, systematic reviews, and meta-analyses performed in Western countries and involving East Asian patients. All of these reported that the route of drug administration, outcome definition, disease severity, disease duration, distal disease, and concomitant corticosteroid treatment influenced outcomes in the placebo arm [62,66,67]. The systematic review and meta-analysis performed by Sedano reported a 32% [95% CI 30–35%; range 6–92%] placebo clinical response rate and 11% [95% CI 9–13%; range 1–49%] placebo clinical remission in induction trials [62].

More variable data were reported by Su in a previous meta-analysis, including a variety of different outcome measures [63]. The reported 0% to 40% remission rates in the placebo arm varied in relation to trial length, the number of study visits, strict remission definition, and disease activity.

For maintenance trials, the pooled placebo clinical response rate is 26% [95% CI 22–31%; range 16–45%] and the pooled placebo clinical remission rate is 18% [95% CI 12–25%; range 5–68%]. Regarding endoscopic response and remission in induction trials, the pooled placebo endoscopic remission rate was 19% [95% CI 15–23%; range 1–68%]. Interestingly, the pooled placebo histological response rate in induction trials was 14% [95% CI 5–33%; range 7–30%] and the pooled placebo histological remission rate was 15% [95% CI 11–19%; range 4–41%]. The same was confirmed in the most recent available meta-analysis, reporting histological remission in 15% patients in induction studies and 14% in maintenance studies [67].

A disease duration exceeding five years prior to enrolment is associated with significantly lower PR rates compared to shorter disease duration [51].

The class of drug tested is also relevant in determining PR and remission rates. Immunosuppressants have the lowest PR and biologics the highest PR (19%, 95% CI 7–43%; *p* = 0.04 vs. 35%, 95% CI 31–38%; *p* < 0.001, respectively) [61,62]. This is likely related to high response expectancy, as biologics are considered the most effective class of medication in UC.

A relevant confounding factor is represented by the concomitant use of corticosteroids or other active drugs in the placebo arm. Corticosteroids are potent anti-inflammatory agents and their concurrent use boosts effects that can be interpreted as PE [61,62]. The standardized use of concomitant medications is essential to ensure the reliability of trial outcomes and the correct evaluation of PR. The exclusion of concomitant active treatment is advisable, when possible. Otherwise, the evaluation of baseline requirements and implementation of tapering schedules ensures minimal variation throughout the trial may prove helpful. The same applies to prior exposure to biological therapies. Patients previously exposed to biologics have higher PR rates in remission maintenance studies due to the extended half-life of the drug and sustained efficacy over time. Thus, the influence of concomitant corticosteroids or other prior therapy on the outcomes of placebo arms in UC trials highlights the need for careful trial design and standardized protocols to minimize confounding factors [61,62]. By addressing these issues, researchers can evaluate the true efficacy of investigational treatments, leading to more reliable and clinically meaningful results.

Considering the administration route, oral agents have a lower PR rate (28%, OR 0.58, 95% CI 0.35 to 0.98) and topically administered agents have a higher PR rate (39%, 95% CI 27–53%) compared to other administration methods [61].

Overall, the timing of the primary outcome assessment is significantly associated with PR rates. Specifically, each one-week increment in the endpoint assessment was linked to an increase in the placebo remission rate. Factors influencing PR and remission rates in UC trials are summarized in Table 2 [59,60,61,62,64,65].

Interestingly, the evaluation of time trends in PR rates showed a steady increase from 13% to 33% between 1987 and 2007, but figures remained stable in the 2008–2015 period (32–34%). Variable PR rates across different classes of drugs and different administration methods were likely responsible for this, underscoring the influence of patient expectation and disease presentation on trial outcomes. The rise in PR rates observed from 1987 to 2007 suggests potential shifts in the composition of patient populations, trial design, or patient perception and response to therapy [61].

## 4. Nocebo Effect

The nocebo effect (NE), derived from the Latin nocere meaning “I shall harm”, refers to a harmful or unpleasant response to a treatment that cannot be attributed to its pharmacodynamic properties, but rather to negative expectations or contextual factors surrounding its administration [6,68]. In this framework, nocebo responses (NRs) represent the negative mirror image of placebo responses, and are triggered by a complex interplay of psychological, social, and neurobiological mechanisms [69]. They may arise due to the patient’s anticipatory anxiety, prior negative treatment experiences, suggestive language used by healthcare professionals, or through social learning—such as exposure to alarming information via media, internet, or peer interactions. NRs can occur in both placebo and active treatment arms, leading to increased rates of perceived adverse events, reduced therapeutic efficacy, and higher dropout rates. From a research standpoint, NR distorts efficacy assessments and may artificially inflate the number of subjects needed to demonstrate the superiority of a new drug in randomized clinical trials [69,70]. Clinically, the nocebo effect undermines trust in medical interventions, contributing to reduced adherence, early discontinuation, and poorer outcomes, particularly in chronic conditions such as UC, which require long-term disease management.

A recent systematic review and meta-analysis of nearly 200 randomized clinical trials, including over 40,000 patients with IBD, reported no statistically significant difference in the rate of serious adverse events between placebo and active treatment arms [71]. Interestingly, while an increased incidence of infections (1–2.9%) was observed among those receiving active therapy, the authors suggest that this may be partially attributed to the nocebo effect, rather than to the immunosuppressive properties of the drugs alone. These findings underscore the potential of negative expectations to magnify perceived or real adverse outcomes, even when biologically unfounded. [72].

The biosimilar switch scenario provides a clear illustration [73]. Although multiple studies have confirmed the safety, efficacy, and immunogenicity equivalence of biosimilars to originator biologics, switch studies in IBD and rheumatology report discontinuation rates of 13% or more, primarily due to subjective complaints or fear of reduced efficacy [68,69]. These effects are not biologically driven but arise from patients’ negative beliefs about cost-effective alternatives, especially when communication is poorly handled or ambiguous [68,74,75]. This is especially relevant considering that biosimilars, despite their similar biological structure and activity, are not exact copies of the original product, but data from the NOR-SWITCH trial confirmed that switching from a biological originator to a biosimilar is non-inferior to continuing with the biological originator [76]. The negative effect, however, is short-lasting and figures decrease from week 16 to week 32, suggesting an initial perception bias that may be reversible through continued positive experience.

Social learning plays a central role in amplifying NR. Observing or hearing about negative experiences, whether in clinic waiting rooms, online forums, or through social media, can help patients to interpret normal bodily sensations as side effects. This phenomenon is particularly relevant in vulnerable groups; female gender, high trait anxiety, and strong empathic capacity are associated with greater nocebo susceptibility due to heightened sensitivity to social cues and suggestive information [77,78,79].

A crucial determinant of NR is physician–patient communication. Framing effects, meaning how information is delivered, can dramatically alter perception. For instance, emphasizing the low probability of side effects rather than listing them in detail can reduce their incidence. Conversely, excessively detailed or negatively framed consent procedures have been shown to increase both the frequency and severity of reported adverse effects [69,80,81,82]. Thus, clinicians must adopt a balanced, honest, but reassuring communication strategy, carefully selecting the language used during patient counseling and the process of obtaining informed consent.

To mitigate NR, a variety of strategies have been proposed (Table 3) [68,69,70,73,80,81,82,83,84]. At the clinical level, the involvement of a multidisciplinary team, including IBD nurses, pharmacists, and behavioral specialists, has been shown to enhance patient support, reduce anxiety, and promote informed yet confident decision-making [69]. IBD nurses play a pivotal role by offering personalized education and continuity of care, bridging gaps in understanding that could otherwise fuel negative expectations.

Additionally, the use of screening tools can help identify patients at higher risk of NR. Instruments such as the Perceived Sensitivity to Medicines (PSM) scale, the Beliefs about Medicines Questionnaire (BMQ), and the Stanford Expectations of Treatment Scale (SETS) have shown potential in anticipating which individuals might require additional support and reframing interventions [84].

In summary, the NE represents a clinically significant and preventable phenomenon, which is particularly relevant in UC where long-term adherence is critical. Understanding its mechanisms and proactively managing communication, expectations, and support can help transform therapeutic relationships and improve both clinical outcomes and patient satisfaction.

## 5. Conclusions

Adherence to therapy is central in determining long-term outcomes and quality of life (QOL) in patients with chronic disease, with UC included. The choice of whether to persist in maintaining therapy depends upon the balance between the need for long-term treatment and the fear of adverse events. Patient beliefs, concerns about possible unwanted effects of drugs, and the interaction with medical staff all result in persistence or discontinuation of therapy. About half of IBD patients willingly accept treatment, but 42% have ambivalent feelings, being convinced of the need for therapy and at the same time concerned about possible adverse events [82]. This large group of patients represents the primary targets of strategies aimed at improving adherence. More so those with high levels of perceived sensitivity to medicines [84] which have been proved to represent a main determinant of nocebo mechanisms, and low persistence, in patients with chronic pain [85]. The same applies to UC patients.

Adherence rates vary in widely different countries, with different healthcare systems, economic situations, and cultural backgrounds. NRs are less likely in relation to male gender, single status, and full-time employment and more likely in the presence of high depression scores and a low quality of life [16]. The issue is age-related and is more complex in adolescents than in childhood or adulthood as patients in this age group are willing to take decisions autonomously [86]. Not surprisingly, adolescents present with the highest therapy discontinuation rate and require special attention.

Non-adherence behavior varies in relation to the class of drug used but also depends on the clinical phase of the disease. This makes the comparison of adherence to different drugs complex as figures vary when data are recorded in activity or remission.

The administration route represents a further critical factor. Although oral therapy and hospital-based injections or infusion are usually the patient’s first choice, hard data are still lacking regarding UC. Topical therapy with foams, suppositories, or enemas is more likely associated with NA than oral therapy [12,61]. Overall, the same applies to the oral route versus parenterally administered therapy. The widely accepted efficacy and high expectancy of first-generation biologics, however, shifted the balance toward parenterally administered treatments, compared to oral mesalamine, in UC [23]. The lack of the immunogenicity of small molecules, reducing the fear of side effects and loss of efficacy, and their oral route of administration favor even better adherence to JAK inhibitors or etrasimod versus parenterally administered first- or second-generation biologics.

The therapeutic gain related to PE is clinically relevant in UC, but data are surprisingly rare outside clinical trials. In this setting, placebo rates vary depending on the assessed endpoint, the trial phase (induction or maintenance), specific trial design features, and the drug family. Subjective assessments are more susceptible to placebo and NE than objective, reproducible criteria. The bias is amplified when the overall evaluation heavily relies on patient-reported outcomes. Normal rectal mucosa on sigmoidoscopy, the limited extent of colitis, and remission—recently achieved by active therapy or maintained for at least 1 year before study entry—increase the PR rate [87]. The histological remission rate was 15.7% in a meta-analysis of UC trials [67]. The interpretation is unclear, as it could depend on the natural history of disease, the effect of background medication, the direct biologic effect of PR, or a combination of them. Understanding the weight of these variables is crucial for designing effective and reliable clinical trials. This will lead to the more accurate assessment of investigational therapies and ultimately improve the management and treatment of UC.

Noteworthy, a rise in PR was recorded from the eighties to the first decade of this millennium. This was followed by a plateau, suggesting changes in the composition of patient populations, trial design, and the patient perception of the efficacy of treatment [61].

Nocebo responses consist in unfavorable treatment outcomes, resulting from negative expectations more than from the side effects of the active ingredient [6,70,73], and are thus minimized by favorable interaction with the attending physician and medical staff [69,70,83]. The close objective monitoring of disease activity using regular biomarkers, endoscopic criteria, and radiological assessment reassure patients, increase confidence, and prevent NE and the discontinuation of therapy [68,83]. From a clinical perspective, gastroenterologists can leverage these findings by adopting several strategies to reduce nocebo effects and reinforce placebo benefits. These actions include the positive framing of treatment benefits [70,85], engaging in shared decision-making to enhance patient trust and agency [69,83], using structured and reassuring informed consent practices [80,81], and providing continuous multidisciplinary support involving nurses and behavioral specialists [68,69]. Screening tools such as the PSM or BMQ can help to identify at-risk individuals early, allowing for tailored communication [84,85]. Reinforcing objective disease monitoring and emphasizing stability or improvement can strengthen patient confidence [83]. Overall, empowering communication, the personalization of care, and minimizing exposure to negative cues are crucial to improving adherence and maximizing outcomes in UC [70,73]. In conclusion, adherence to maintenance therapy in UC is of prime importance in reducing the relapse rate, improving the QOL of patients and possibly reducing the risk of cancer. A complex interplay of differing factors modulates the likelihood of persisting or discontinuing treatment. Patient’s psychosocial factors, disease severity and course, the drug(s) used, and the interaction with medical personnel all concur with the result, as does the PE/NE.

Attention has been mainly given to PE in the setting of clinical trials, where it represents an important source of bias. Efforts are made to minimize its occurrence in order to establish the true efficacy of the intervention under evaluation, thereby enabling more robust and efficient study designs. The opposite applies to the normal day-to-day clinical setting in which patient–physician/medical staff communication triggers positive effects. Reassurance and a positive approach favorably affect therapy persistence and the eventual outcomes. Moreover, the same attitude potentially boosts PR and may decrease functional symptoms in patients with inactive and minimally active disease that nonetheless present with abdominal discomfort. It is noteworthy that the effects of positive or negative expectations result in PE or NE over time. Thus, the best approach and interaction strategies should vary accordingly. A patient’s confidence in their ability to manage disease-related demands in IBD can be evaluated through a validated questionnaire [88]. The stratification of patients according to this feature may help to actively modify patients’ psychological characteristics and optimize the physician/patient interaction.

Improved adherence, outcomes, and QOL are favored by an individualized approach, but overall, increasing the frequency of patient visits enhances the perception of effective care and attention, and favors good adherence to prescribed therapy. Conversely, suboptimal communication with medical personnel may result in a focus on possible AE rather than the efficacy of therapies. Unintentionally delivered negative expectations increase the rate of NE and intentional non-adherence.

What the patient thinks and his attitude toward treatment reflects on what the patient does, amplifying or reducing the placebo–nocebo response and the likelihood of persistent, effective long-term maintenance therapy. Knowledge of the mechanisms underlying patients’ behavior may help identifying which therapeutic approach and which treatment schedule any individual patient is more likely to accept. It also improves strategies aimed at optimizing adherence, outcomes, and QOL.

## Figures and Tables

**Table 1 jcm-14-04351-t001:** General features in clinical trials that minimize or boost placebo response.

Key Factor/Feature	Description	References
Protocol Design	Patient enrolment, assessment and follow-up are critical.	Enck 2013 [59]
Endpoint	Less PE with objective measurement of disease activity (e.g., endoscopic scores) than with patient-reported outcomes.	Enck 2013 [59]; Wong 2023 [60]; Jairath 2017 [61]
Screening	Less PE with strict inclusion and exclusion criteria selecting homogeneous patient populations.	Jairath 2017 [61]
Disease Severity	Less PR and remission rates in patients with higher endoscopic sub-scores at enrolment and prior exposure to biologics.	Wong 2023 [60]; Jairath 2017 [61]
Central Reading of Disease Activity Assessment	Decreased PR rate ensures objective and reproducible assessment across all trial sites.	Jairath 2017 [61]
Concomitant Therapy	Standardization minimizes PE rates. Tapering protocols for steroids is crucial.	Jairath 2017 [61]; Sedano 2022 [62]
Data Analysis	The subgroup analyses help defining the impact of PE on trial outcomes.	Jairath 2017 [61]
Training and Monitoring of Investigators	Strict adherence to trial protocol, regular monitoring and audits minimizes interfering factors.	Enck 2013 [59]
Timing of Primary Endpoint Measurement *	PE is reduced by shorter visit intervals.	Jairath 2017 [61]
Follow-Up *	PE is reduced by limited number of follow-up visits. This minimizes patient expectations favored by patient-staff interaction.	Enck 2013 [59]
Longer Trial Duration *	PE is reduced in long- duration trials.	Enck 2013 [59]; Su 2007 [63]

The three features identified by * are partially conflicting. PE = placebo effect, PR = placebo response.

**Table 2 jcm-14-04351-t002:** Factors influencing placebo response and remission rates in UC trials.

Category	Content	References
Drug Class	Biological drugs: highest PR rates. Immunosuppressants: lowest PR rates. Aminosalicylates: highest placebo remission rates.	Jairath 2017 [61]; Sedano 2022 [62]
Administration Method	Oral Agents: lower PR rates than topical agents.	Jairath 2017 [61]
Disease Severity at Enrollment	High endoscopic subscore lowers PR and remission rates. Mild disease associated with higher PR.	Wong 2023 [60]; Jairath 2017 [61]
Patient History	Low PR rates associated with prior exposure to biologics and long disease duration (>5 years).	Jairath 2017 [61]
Eligibility Criteria	PR rate increased when minimum rectal bleeding score is required, compared to trials who do not.	Enck 2013 [59]; Wong 2023 [60]
Trial Design Features	Employ standardized disease activity indices for patient enrolment and assessment. Placebo endoscopic response and remission rates decreased by central reading. Higher PR is associated with frequent visits. High PR rates associated with longer intervals to primary endpoint assessment.	Enck 2013 [59]; Jairath 2017 [61]; Sedano 2022 [62]; Sutherland 1987 [64]; Schroeder 1987 [65]

UC = ulcerative colitis, PR = placebo response.

**Table 3 jcm-14-04351-t003:** Strategies to reduce the nocebo effect in clinical trials and practice.

Key Strategy	Implementation Details	References
Positive Framing of Information	Emphasize benefits and low risk of side effects instead of detailing every possible adverse event.	D’Amico 2021 [68], Colloca 2020 [70], Colloca 2019 [73]
Tailored Informed Consent	Adapt the depth and tone of consent to patient needs while remaining transparent and balanced.	Colloca 2020 [70], Mondaini 2007 [80], Haas 2022 [81], Pouillon 2019 [83]
Empathic Patient Communication	Maintain trust with empathetic, open dialogue to reduce anxiety and expectation of harm.	D’Amico 2021 [68], Colloca 2020 [70], Colloca 2019 [73], Pouillon 2019 [83]
Training of Healthcare Staff	Educate staff to avoid unintentional cues or language that might trigger nocebo responses.	D’Amico 2020 [69], Pouillon 2019 [83]
Patient Risk Screening	Use tools like PSM, BMQ, or SETS to identify patients with high sensitivity or negative expectations.	Horne 2009 [82], Horne 2013 [84]
Educational Interventions	Provide clear, supportive explanations about treatment mechanisms, biosimilars, and expected outcomes.	D’Amico 2021 [68], Colloca 2019 [73], Wetwittayakhlang 2024 [75], Pouillon 2019 [83]
Shared Decision-Making	Involve patients in treatment choices to empower them and reduce perceived lack of control.	D’Amico 2020 [69], Pouillon 2019 [83]
Use of Telehealth Follow-Up	Reinforce trust and consistency in care with scheduled, supportive remote contact.	D’Amico 2021 [68]
Addressing Social Learning	Correct misinformation and mitigate anxiety from peer influence or online sources.	Rooney 2023 [77], Saunders 2024 [78], Faasse 2019 [79]
Multidisciplinary Support	Engage nurses, psychologists, and pharmacists to reinforce consistent, coordinated care.	D’Amico 2021 [68], D’Amico 2020 [69]

## Data Availability

The original contributions presented in this study are included in the article. Further inquiries can be directed to the corresponding authors.

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
