# Peer review of "What the Patient Thinks and What the Patient Does: Placebo, Nocebo, and Therapy Adherence in Ulcerative Colitis"

_jcm, 2025, doi:10.3390/jcm14124351_

Round 1

Reviewer 1 Report

Comments and Suggestions for Authors

Thank you for the opportunity to review the manuscript entitled "What the patient thinks and what the patient does. Placebo, 2 nocebo and adherence to therapy in ulcerative colitis".

The idea of the review is interesting, but I think the publications should be thoroughly rewritten. For example: the introduction lacks a description of the disease entity, a brief introduction to treatment. Although there is nocebo in the title, only a small part is described in the manuscript.

Nevertheless, I wish the authors all the best and keep my fingers crossed for the expansion of the manuscript.

Author Response

Thank you for the opportunity to review the manuscript entitled "What the patient thinks and what the patient does. Placebo, 2 nocebo and adherence to therapy in ulcerative colitis".

The idea of the review is interesting, but I think the publications should be thoroughly rewritten. For example: the introduction lacks a description of the disease entity, a brief introduction to treatment.

We appreciate the Reviewer’s suggestion. We have amplified the Introduction by including a brief overview of ulcerative colitis (UC) as a chronic inflammatory disease, along with a concise description of available treatment strategies (lines 28–48).
Additionally, the manuscript has undergone a comprehensive English language revision to improve fluency, ensure grammatical accuracy, and enhance overall readability throughout the text.

 Although there is nocebo in the title, only a small part is described in the manuscript. Nevertheless, I wish the authors all the best and keep my fingers crossed for the expansion of the manuscript

In response, we have substantially expanded the section on the nocebo effect (Chapter 4). The revised chapter provides a more comprehensive overview of the mechanisms, clinical impact, and contributing psychosocial factors of the nocebo effect, particularly in the context of UC. Line 371-435. 

We also added a new summary table (Table 3), which outlines evidence-based strategies to mitigate nocebo responses in clinical practice. This addition aims to offer practical tools for healthcare providers and is supported by updated references [68, 69, 73, 76].

Reviewer 2 Report

Comments and Suggestions for Authors

The current review article provides an interesting analysis of patients' beliefs and behaviors related to medications prescribed for their ulcerative colitis, and how these beliefs and behaviors impact their health outcomes. Overall, this is a well-organized review on a clinically meaningful topic, and could be further improved with English language editing and additional information and sources to provide more context.

Because medication adherence depends on personal/cultural factors, it would be useful to discuss characteristics of the sample populations used in the cited studies, e.g., what country the study was conducted in, rural vs. urban setting, socialized medical system vs. other system, etc. This is particularly relevant to the use of complementary/alternative medicine. Similarly, it's pertinent to note how NA data was gathered in the studies, i.e., self-report (which might under-estimate NA) or pharmacy fill history. The paper would also benefit from conclusions on how gastroenterologists can use the data on placebo/nocebo effects to provide better clinical care.

Comments on the Quality of English Language

There are some spelling and grammar errors that are noticeable to native English speakers, including but not limited to:

  • "believes" instead of "beliefs"  
  • Abbreviations should be defined before they are used, e.g. "ulcerative colitis (UC)"
  • "Countries" does not need to be capitalized
  • Occasional incorrect prepositions and comma placement
  • "Mediterranean" should be capitalized
  • "PA" instead of "A" at the start of 2.4

Author Response

The current review article provides an interesting analysis of patients' beliefs and behaviors related to medications prescribed for their ulcerative colitis, and how these beliefs and behaviors impact their health outcomes. Overall, this is a well-organized review on a clinically meaningful topic, and could be further improved with English language editing and additional information and sources to provide more context.

We appreciate the Reviewer’s constructive feedback. The entire manuscript has undergone careful English language editing to improve clarity, grammar, and readability.

We have also addressed the request for greater contextual depth by expanding several key sections of the manuscript, including:

  • Introduction: a brief overview of ulcerative colitis (UC) as a chronic inflammatory disease, along with a concise description of available treatment strategies has been added (lines 28–88).
  • Chapter on adherence: We have incorporated a more detailed information about critical factors influencing adherence, such as healthcare system differences, cultural attitudes, and geographical settings, particularly in the sections addressing complementary and alternative medicine (CAM), variability in medication adherence, and methods used to assess non-adherence. Lines 79-82; 92-97; 217-222.
  • Chapter on the nocebo effect. The revised chapter provides a more comprehensive overview of the mechanisms, clinical impact, and contributing psychosocial factors of the nocebo effect, particularly in the context of ulcerative colitis.

We also added a new summary table (Table 3), which outlines evidence-based strategies to mitigate nocebo responses in clinical practice. This addition aims to offer practical tools for healthcare providers and is supported by updated references [68, 69, 73, 76]. Line 371-435.  

Because medication adherence depends on personal/cultural factors, it would be useful to discuss characteristics of the sample populations used in the cited studies, e.g., what country the study was conducted in, rural vs. urban setting, socialized medical system vs. other system, etc. This is particularly relevant to the use of complementary/alternative medicine. Similarly, it's pertinent to note how NA data was gathered in the studies, i.e., self-report (which might under-estimate NA) or pharmacy fill history.

Thank you for this insightful point. We have clarified the methods used to assess non-adherence (NA) in several key studies, specifying whether they relied on self-reports, pharmacy refill data, or therapeutic drug monitoring. Additional context on healthcare systems, cultural factors, and geographic settings has also been added, particularly in sections on CAM, adherence variability, and non-adherence assessment (lines 79–82, 92–97, 217–222).

The paper would also benefit from conclusions on how gastroenterologists can use the data on placebo/nocebo effects to provide better clinical care.

We agree with the Reviewer. We have expanded the conclusions to include practical, evidence-based recommendations for clinicians, including framing strategies to minimize nocebo responses, tools for identifying at-risk patients, and empowering communication approaches to enhance placebo-related benefits. Additionally, we have introduced Table 3, which summarizes these actionable strategies with supporting references.

Comments on the Quality of English Language

There are some spelling and grammar errors that are noticeable to native English speakers, including but not limited to:

  • "believes" instead of "beliefs"  
  • Abbreviations should be defined before they are used, e.g. "ulcerative colitis (UC)"
  • "Countries" does not need to be capitalized
  • Occasional incorrect prepositions and comma placement
  • "Mediterranean" should be capitalized
  • "PA" instead of "A" at the start of 2.4

We thank the Reviewer for highlighting this. The manuscript has undergone extensive English language revision, and all noticeable spelling, grammar, and syntax issues have been carefully corrected.

Reviewer 3 Report

Comments and Suggestions for Authors

The article provides an overview and summary of important information for doctors and patients with ulcerative colitis. This information concerns very significant phenomena such as the placebo and nocebo effects, as well as patient behavior in specific clinical situations. I believe this information is very important and, although known, it is rarely addressed in the literature. 

Author Response

The article provides an overview and summary of important information for doctors and patients with ulcerative colitis. This information concerns very significant phenomena such as the placebo and nocebo effects, as well as patient behavior in specific clinical situations. I believe this information is very important and, although known, it is rarely addressed in the literature. 

We sincerely thank the reviewer for the positive and encouraging feedback.
We greatly appreciate your recognition of the relevance of placebo and nocebo effects, as well as patient behavior in ulcerative colitis, topics that, although known, are indeed underexplored in clinical literature. Our aim was precisely to provide a clinically meaningful and practical overview to support both physicians and patients in managing long-term therapy more effectively.

Your comment reinforces the importance of highlighting these concepts and encourages us in continuing to address often-overlooked aspects of patient-centered care.